# A Multivariable Mendelian Randomization Study of Systolic and Diastolic Blood Pressure, Lipid Profile, and Heart Failure Subtypes

**DOI:** 10.3390/genes15091126

**Published:** 2024-08-27

**Authors:** Chang Liu, Qin Hui, Quinn S. Wells, Eric Farber-Eger, John Michael Gaziano, Peter W. F. Wilson, Arshed A. Quyyumi, Viola Vaccarino, Yi-Juan Hu, David Benkeser, Lawrence S. Phillips, Jacob Joseph, Yan V. Sun

**Affiliations:** 1Department of Epidemiology, Rollins School of Public Health, Emory University, Atlanta, GA 30322, USA; chang.liu2@emory.edu (C.L.); qhui@emory.edu (Q.H.); pwwilso@emory.edu (P.W.F.W.); lvaccar@emory.edu (V.V.); 2Atlanta VA Healthcare System, Decatur, GA 30033, USA; medlsp@emory.edu; 3Division of Cardiovascular Medicine, Department of Medicine, Vanderbilt University, Nashville, TN 37232, USA; quinn.s.wells@vumc.org (Q.S.W.); eric.h.farber-eger@vumc.org (E.F.-E.); 4Division of Aging, Department of Medicine, Brigham and Women’s Hospital, Harvard Medical School, Boston, MA 02115, USA; jmgaziano@bwh.harvard.edu; 5Massachusetts Veterans Epidemiology Research and Information Center (MAVERIC), VA Boston Healthcare System, Boston, MA 02111, USA; 6School of Medicine, Emory University, Atlanta, GA 30322, USA; aquyyum@emory.edu; 7Department of Biostatistics and Bioinformatics, Rollins School of Public Health, Emory University, Atlanta, GA 30322, USA; yijuan.hu@emory.edu (Y.-J.H.); benkeser@emory.edu (D.B.); 8VA Providence Healthcare System, Providence, RI 02908, USA; jacob.joseph@va.gov; 9The Warren Alpert Medical School, Brown University, Providence, RI 02903, USA

**Keywords:** heart failure, HFrEF, HFpEF, Mendelian randomization, blood pressure, lipids

## Abstract

Heart failure (HF) is a significant health burden, with two major clinical subtypes: HF with reduced (HFrEF) and preserved ejection fraction (HFpEF). Blood pressure and lipid profile are established risk factors of HF. We performed univariable and multivariable Mendelian randomization (MR) analyses to assess potential causal effects of blood pressures and lipids on HF subtypes. Genetic instruments for blood pressures and lipids were derived from genome-wide association studies (GWASs) among the European participants of the UK Biobank. GWAS summaries of HFrEF and HFpEF were obtained from the meta-analysis of the European participants from the Million Veteran Program and the Vanderbilt University DNA Databank. Systolic blood pressure exhibited a supportive MR association primarily with HFpEF (odds ratio [OR], 1.14; 95% confidence interval [CI], 1.04–1.23), while diastolic blood pressure had an independent MR association with HFrEF (OR, 1.43; 95% CI, 1.13–1.77). MR associations also supported the observation that higher levels of low-density lipoprotein cholesterol increase the risk for both subtypes (HFrEF OR, 1.10 and 95% CI, 1.05–1.17; HFpEF OR, 1.05 and 95% CI, 1.02–1.09). These findings underscore differences in HF subtype-specific risk profiles and mechanisms, which may lead to different interventional strategies for different HF subtypes.

## 1. Introduction

Heart failure (HF) impacts approximately 6.2 million individuals in the United States [1]. It is linked to substantial morbidity, mortality, and healthcare expenses, surpassing USD 31 billion in 2020 [2]. The prevalence of HF in the U.S. has been on a steady rise, with estimates projecting a total of over 8 million cases and healthcare costs exceeding USD 53 billion by 2030 [1,2]. Clinical subcategories are delineated by the left ventricular ejection fraction (LVEF), with two major subtypes that account for approximately 90% of all heart failure (HF) cases. These subtypes are HF with reduced ejection fraction (HFrEF), characterized by an LVEF ≤ 40%, and HF with preserved ejection fraction (HFpEF), identified by an LVEF ≥ 50% [3]. HFrEF is primarily attributed to myocardial remodeling and contractile dysfunction, whereas HFpEF is characterized by compromised left ventricular filling [4]. Due to the distinct pathophysiological underpinnings of HFrEF and HFpEF, recent investigations, including clinical trials, examine these subtypes individually.

The risk of HF is influenced by genetic factors, as indicated by an estimated heritability of 26% [5]. The genetic architecture of HFrEF and HFpEF can be different given their distinct pathophysiology and risk profile. A recent large-scale genome-wide association study (GWAS) of HF subtypes identified 13 genomic loci associated with HFrEF and 1 locus associated with HFpEF [6]. Separate GWASs for HFrEF and HFpEF have enhanced the efficiency of pinpointing genetic predispositions and uncovering the genetic mechanisms driving HF.

Both blood pressure and lipid profile are established risk factors of HF, and hypertension accounts for an estimated 17%–28% of HF cases and dyslipidemia (either elevated non-HDL-C or low HDL-C) accounts for 22.5% of HF cases [7,8]. A recent large study of the Million Veteran Program (MVP), which included over 1.9 million individuals, estimated the quantitative differences in the magnitude of association between blood pressure, lipids, and incident HF subtypes and showed that systolic blood pressure is more strongly associated with HFpEF than with HFrEF [9]. However, the causal effects of blood pressure and lipids on HF subtypes, particularly their independent effects, have not been evaluated. It has been shown that intensive blood pressure lowering can effectively prevent HF [10], and similarly, statin therapy has been shown to be effective in reducing the risk of nonfatal HF hospitalization [11]. Growing evidence has indicated a causal effect of blood pressure and lipids in HF and its subtypes. In addition, blood pressure measures are associated with lipid profile components, with successful replications across multiple populations [12], which highlights the importance of dissecting the independent causal effect of the subcomponents of blood pressure and lipid profile on HF subtypes.

In this context, our study aims to provide a comprehensive evaluation of the independent causal effects of blood pressure and lipid profiles on the risk of HFrEF and HFpEF. Specifically, we utilize a multivariable two-sample Mendelian randomization (MR) design to investigate the causal relationships between HFrEF and HFpEF and systolic blood pressure (SBP), diastolic blood pressure (DBP), pulse pressure (PP), high-density lipoprotein cholesterol (HDL-C), low-density lipoprotein cholesterol (LDL-C), and triglycerides. Our hypothesis is that these correlated factors may exert independent causal effects on HF subtypes, potentially differing in magnitude due to the distinct pathophysiological mechanisms underlying HFrEF and HFpEF. By elucidating these causal relationships, we aim to enhance the understanding of HF etiology and contribute to more targeted preventive and interventional strategies through common risk factors of HF. This is particularly important for HFpEF, which has very limited treatment options.

## 2. Materials and Methods

To identify genetic instrumental variables for SBP, DBP, PP, HDL-C, LDL-C, and triglycerides, we first performed a genome-wide association study (GWAS) for each trait among the UK Biobank participants of European ancestry after excluding individuals with 3rd or higher degrees of relatedness [13]. The specific sample size included in the GWAS for each trait is listed in Appendix A. The GWAS was performed under the assumption of the additive genetic model using linear regression models, adjusting for age, sex, blood pressure medication for SBP and DBP, cholesterol-lowering medication for lipids, and the top 10 principal components, which were derived from the genotyping data after linkage disequilibrium (LD) pruning, with an r^2^ threshold of 0.1 within a 250-SNP window and a step of 5 SNPs. A total of 9,787,687 SNPs with a minor allele frequency >1% were tested.

For each GWAS summary, we employed the FUMA v1.3.5d [14] platform to delineate independent genome-wide significant (GWS) loci with *p* < 5 × 10^−8^. Based on the LD structure, independent SNPs were defined at an r^2^ of 0.6, followed by secondary significant SNPs discerned at an r^2^ of 0.1. Loci-spanning distances of less than 250 kb were integrated, with the most significant SNP designated as the sentinel SNP for each locus. The resulting sentinel GWS SNPs excluding indels were used as genetic instrumental variables for downstream MR analyses.

The GWAS summaries for HFrEF and HFpEF were based on the large-scale meta-analysis among participants of European ancestry from the MVP [15] and the Vanderbilt University DNA Databank (BioVU) [16]. The MVP is a large national mega-biobank that has recruited veterans from over 60 Veterans Health Administration medical centers nationwide since 2011 [15]. The BioVU is a biorepository of de-identified samples collected from discarded blood after routine clinical testing [16]. In both cohorts, HF subtype phenotyping was performed based on a combination of International Classification of Diseases (ICD) codes including 428.x for ICD-9 and I50.x for ICD-10 and echocardiograms within 6 months of diagnosis, with subclassifications including HFrEF cases (LVEF ≤ 40%) and HFpEF cases (LVEF ≥ 50%); controls were participants with no recorded HF diagnosis [6]. The MVP cohort included 27,799 HFrEF cases, 27,579 HFpEF cases, and 367,267 controls; the BioVU cohort included 1263 HFrEF cases, 1702 HFpEF cases, and 12,783 controls. Assuming an additive genetic model, each GWAS by cohort and HF subtype was performed using logistic regression models while accounting for age, sex, and the top 10 principal components. Subsequently, the meta-analysis was conducted using the Genome-Wide Association Meta-Analysis (GWAMA v2.2.2) software [17], which incorporated a random-effects model, accommodating variations in effect sizes due to heterogeneity.

We first conducted univariable two-sample MR analyses to estimate the individual effect of blood pressure and lipids using the R package TwoSampleMR (v0.6.4), which incorporated multiple MR models, including an inverse-variance weighted model and the MR Egger regression model to ensure an accurate causal estimate [18]. The independent genetic loci for SBP, DBP, PP, HDL-C, LDL-C, and triglycerides were utilized as genetic instruments for the univariable MR analysis. For each trait, we exclusively selected genetic instruments that did not exhibit genome-wide significance (*p* < 5 × 10^−8^) in the analysis of other traits. This approach was taken to reduce the potential impact of pleiotropy. Steiger filtering was applied to exclude genetic variants that have a stronger association with the outcome than exposure, as a protection of reverse causation [19]. Egger’s test for the intercept term in the Egger regression model was used for the testing of the no pleiotropy assumption [20]. As an additional layer of model curation, the Mendelian Randomization Pleiotropy RESidual Sum and Outlier (MR-PRESSO) test [21] was conducted to further evaluate pleiotropy. Further, the independent causal effects of SBP, DBP, HDL-C, LDL-C, and triglycerides were estimated using the multivariable MR (MVMR) design [22], which collectively uses the corresponding genetic instruments. PP was not included in this analysis because it was directly calculated as the difference between SBP and DBP. All statistical analyses were performed using R version 4.2.1 (https://www.R-project.org/ (accessed on 24 April 2024)).

## 3. Results

In the univariable MR analysis, every 10 mmHg rise in SBP was associated with higher risks of HFrEF and HFpEF based on the IVW model after the removal of outliers, with the MR-Egger model demonstrating no significant pleiotropy (OR, 1.19; 95% CI, 1.06–1.34; *p* = 3.31 × 10^−3^ and OR, 1.19; 95% CI, 1.09–1.30; *p* = 1.12 × 10^−4^, respectively). DBP showed a potential effect on HFrEF (OR, 1.47; 95% CI, 1.28–1.70; *p* = 7.95 × 10^−8^) but only a marginal effect on HFpEF, with confidence intervals covering the null. PP showed effects on both HFrEF and HFpEF (OR, 1.20; 95% CI, 1.09–1.31; *p* = 9.85 × 10^−5^ and OR, 1.17; 95% CI, 1.09–1.26; *p* = 1.81 × 10^−5^, respectively), as shown in Table 1. There was no discernible effect of HDL-C levels on either subtype of HF. Higher levels of LDL-C showed hazardous effects (OR, 1.05; 95% CI, 1.03–1.07; *p* = 4.13 × 10^−5^ and OR, 1.04; 95% CI, 1.01–1.06; *p* = 1.28 × 10^−3^, respectively, for HFrEF and HFpEF), and every 10% rise in triglyceride values showed a moderately increased risk of HFrEF (OR, 1.02; 95% CI, 1.00–1.03; *p* = 0.01), despite the remaining pleiotropy after the removal of outliers using MR-PRESSO, as shown in Table 1. The numbers of genome-wide significant loci related to blood pressure and lipids are shown in Appendix A. All GWAS summaries used for the genetic instruments are listed in Appendix A.

In the multivariable MR analysis, every 10 mmHg increase in SBP exhibited an effect on HFpEF (OR, 1.14; 95% CI, 1.04–1.23) but not on HFrEF; DBP showed an effect on HFrEF (OR, 1.43; 95% CI, 1.13–1.77) but only a marginal effect on HFpEF (OR, 1.05; 95% CI, 0.90–1.19), as shown in Table 2 and Figure 1. HDL-C showed a marginal protective effect on HFpEF (OR, 0.96; 95% CI, 0.90–1.01), with the confidence interval covering the null. LDL-C demonstrated a higher risk in both HFrEF (OR, 1.10; 95% CI, 1.05–1.17) and HFpEF (OR, 1.05; 95% CI, 1.02–1.09). There was no discernible effect of triglyceride levels on either subtype of HF, as shown in Table 2 and Figure 1.

## 4. Discussion

This study used the MR framework to evaluate the independent causal effects of blood pressure and lipid profiles on HF subtypes. Earlier inquiries had mostly concentrated on mechanisms and potential druggable targets of HF as a composite phenotype [23,24,25,26,27], a limitation attributed to the absence of extensive GWASs of HF subtypes that are necessary for identifying reliable genetic instrumental variables. A recent MR study on HFrEF and HFpEF, which was conducted following the first GWAS on HF subtypes, relied solely on a univariable MR framework [6]. Our study builds on this by employing multivariable MR analyses, which disentangle the independent and distinct roles of blood pressure and lipid profiles in the etiology of HF subtypes. The differences observed between univariable and multivariable MR results underscore the importance of accounting for potential pleiotropic effects that may be missed in traditional univariable MR analyses.

Notably, an increase in SBP showed a potential causal effect on HFpEF but not on HFrEF, which is consistent with previous longitudinal studies showing a stronger association between SBP and HFpEF than between SBP and HFrEF [9]. SBP primarily reflects the pressure the heart works against during systole, which directly influences the stress and strain on the left ventricle. In contrast, DBP reflects the pressure during diastole, which influences the heart’s filling pressure and ventricular preload. The most common cause of HFpEF is left ventricular diastolic dysfunction, caused by longstanding hypertension, especially elevated SBP, which leads to hypertrophy of the left ventricle [28]. The chronic pressure overload causes the left ventricle to increase in muscle mass, and this can lead to myocardial ischemia, which occurs when the heart muscle does not receive enough blood flow. The resulting fibrosis and stiffness of the left ventricle can further impair its ability to be filled with blood [29].

In clinical trials of hypertension treatments, the relationship between reductions in left ventricular mass and changes in blood pressure values remains contentious. In studies by Rzeznik et al. and Zeller et al. [30,31], left ventricular mass reduction induced by renal artery stenting or percutaneous transluminal renal angioplasty was independent of the change in blood pressure, whereas a study by Yoshitomi et al. [32] reported correlations between the changes in mean blood pressure and left ventricular mass index (r = 0.77, *p* < 0.01) in patients with primary aldosteronism and unilateral renovascular hypertension after treatment. Interestingly, another trial of an antihypertensive treatment by Rowlands et al. [33] suggested that SBP has a stronger correlation with left ventricular mass compared to DBP. In addition, other studies have demonstrated that both left ventricular mass and SBP decrease in parallel during antihypertensive treatment in the context of left ventricular hypertrophy [34,35]. Collectively, the growing evidence suggests that SBP may play a causal role in the development of HFpEF through its impact on increasing left ventricular mass. Furthermore, hormonal mechanisms, such as those involving aldosterone, may contribute to inflammation, fibrosis, and stiffness of the heart [36,37]. Conversely, DBP showed a significant association with MR only in HFrEF, which may have been due to its association with an increased risk of CAD and myocardial infarction, a known contributor to left ventricular contractile dysfunction and HFrEF. These distinctions highlight the need for a nuanced understanding of the contributions of blood pressure components and specific treatment regimens to treat different HF subtypes, with a focus on blood circulation through the heart.

Regarding lipid levels, our results indicated that higher levels of HDL-C exhibited a marginal protective effect in HFpEF and possibly emphasize the role of HDL-C on inflammation and impaired endothelial functions [38], which are common factors in HFpEF [39]. In contrast, higher levels of LDL-C were associated with an elevated risk of both HFrEF and HFpEF, with a larger effect size for HFrEF, emphasizing the role of LDL-C on atherosclerotic cardiovascular disease, which is more common in the HFrEF subtype [40]. Interestingly, despite the established role of lipids in cardiovascular health, our analysis did not find any discernible effect of triglyceride levels on either HF subtype. This suggests that the association between triglycerides and HF risk may not be as prominent while controlling for the potential causal effects of lipoproteins. This aligns with a previous MVMR study on coronary artery disease, which showed that triglycerides on their own did not directly cause coronary artery disease, and as a risk factor, their association is often attributed to the multifaceted impacts and pleiotropic effects linked to genetic variations in LDL-C and HDL-C SNPs [41]. Furthermore, a recent clinical trial on the use of pemafibrate, a medication that lowers triglyceride levels, did not result in a reduction in the risk of cardiovascular events [42].

While this MR study provides novel insights into HF subtype etiology, certain limitations warrant consideration. The reliance on genetic instrumental variables assumes the validity of the genetic variants, which act as proxies for the risk factors. Additionally, the study’s findings are based on aggregated genetic information from European populations, potentially limiting the generalizability to other specific demographic populations. Despite these limitations, this study’s approach offers valuable insights into the mechanisms of HF subtypes, prompting further research and contributing to a broader understanding of cardiovascular health.

## 5. Conclusions

This in-depth MR analysis provides novel insights into the distinct effects of blood pressure components and lipid levels on HF subtypes. These findings underscore the importance of considering subtype-specific mechanisms in understanding HF etiology. While this study advances our understanding, further research is needed to unravel the complex interactions between these factors and their contributions to different HF subtypes.

## Figures and Tables

**Figure 1 genes-15-01126-f001:**
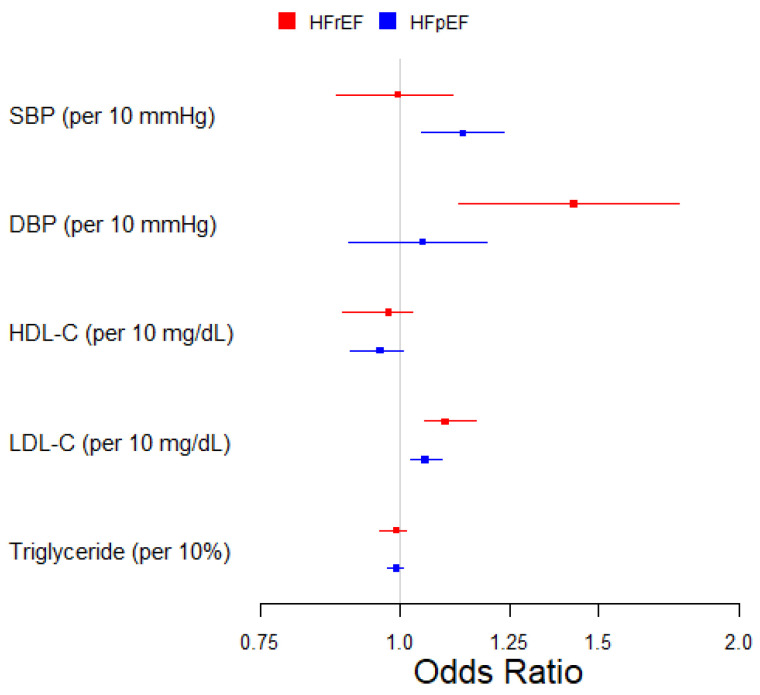
Multivariable two-sample Mendelian randomization of the association between blood pressure, lipids, and HF subtypes.

**Table 1 genes-15-01126-t001:** Univariable two-sample Mendelian randomization of the association between blood pressure, lipids, and HF.

Exposure	Model	HFrEF				HFpEF			
		N GIVs	OR (95% CI)	*p*	Pleiotropy Test *p*	N GIVs	OR (95% CI)	*p*	Pleiotropy Test *p*
SBP (per 10 mmHg)	MR Egger	75	0.68 (0.41, 1.13)	0.14	**0.03**	74	1.17 (0.81, 1.69)	0.41	1.00
	IVW		**1.17 (1.02, 1.34)**	**0.03**			**1.17 (1.07, 1.28)**	**9.24 × 10^−4^**	
	MR Egger (MR-PRESSO)	71	0.81 (0.52, 1.26)	0.35	0.08	73	1.25 (0.88, 1.76)	0.22	0.79
	IVW (MR-PRESSO)		**1.19 (1.06, 1.34)**	**3.31 × 10^−3^**			**1.19 (1.09, 1.3)**	**1.12 × 10^−4^**	
DBP (per 10 mmHg)	MR Egger	83	1.85 (0.99, 3.47)	0.06	0.52	84	0.97 (0.55, 1.69)	0.90	0.73
	IVW		**1.51 (1.29, 1.77)**	**2.32 × 10^−7^**			1.06 (0.92, 1.22)	0.40	
	MR Egger (MR-PRESSO)	80	**1.88 (1.07, 3.32)**	**0.03**	0.39	84	0.97 (0.55, 1.69)	0.90	0.73
	IVW (MR-PRESSO)		**1.47 (1.28, 1.7)**	**7.95 × 10^−8^**			1.06 (0.92, 1.22)	0.40	
PP (per 10 mmHg)	MR Egger	145	0.98 (0.72, 1.34)	0.91	0.30	146	**1.29 (1.01, 1.65)**	**0.04**	0.41
	IVW		**1.15 (1.03, 1.27)**	**8.96 × 10^−3^**			**1.17 (1.08, 1.27)**	**1.27 × 10^−4^**	
	MR Egger (MR-PRESSO)	141	1.02 (0.78, 1.34)	0.86	0.23	142	**1.37 (1.1, 1.7)**	**5.95 × 10^−3^**	0.15
	IVW (MR-PRESSO)		**1.2 (1.09, 1.31)**	**9.85 × 10^−5^**			**1.17 (1.09, 1.26)**	**1.81 × 10^−5^**	
HDL-C (per 10 mg/dL)	MR Egger	149	1.04 (0.89, 1.22)	0.59	0.25	149	1.1 (0.96, 1.25)	0.18	0.06
	IVW		0.96 (0.89, 1.04)	0.33			0.98 (0.92, 1.05)	0.56	
	MR Egger (MR-PRESSO)	144	1.07 (0.94, 1.22)	0.33	0.10	148	1.1 (0.96, 1.25)	0.16	0.07
	IVW (MR-PRESSO)		0.97 (0.91, 1.03)	0.34			0.99 (0.93, 1.05)	0.71	
LDL-C (per 10 mg/dL)	MR Egger	142	**1.1 (1.05, 1.14)**	**4.08 × 10^−5^**	0.09	142	**1.05 (1.01, 1.09)**	**0.01**	0.38
	IVW		**1.06 (1.04, 1.09)**	**1.96 × 10^−6^**			**1.04 (1.01, 1.06)**	**2.49 × 10^−3^**	
	MR Egger (MR-PRESSO)	138	**1.07 (1.03, 1.11)**	**9.59 × 10^−4^**	0.25	141	**1.05 (1.01, 1.09)**	**0.01**	0.48
	IVW (MR-PRESSO)		**1.05 (1.03, 1.07)**	**4.13 × 10^−5^**			**1.04 (1.01, 1.06)**	**1.28 × 10^−3^**	
Triglycerides (per 10%)	MR Egger	151	0.99 (0.97, 1.02)	0.55	**0.02**	151	0.99 (0.97, 1.01)	0.52	**0.03**
	IVW		**1.02 (1, 1.03)**	**0.04**			1.01 (1, 1.03)	0.09	
	MR Egger (MR-PRESSO)	147	1 (0.97, 1.02)	0.75	**0.02**	148	1 (0.98, 1.02)	0.75	0.06
	IVW (MR-PRESSO)		**1.02 (1, 1.03)**	**0.01**			1.01 (1, 1.03)	0.06	

SBP: systolic blood pressure; DBP: diastolic blood pressure; HDL-C: high-density lipoprotein cholesterol; LDL-C: low-density lipoprotein cholesterol; GIV: genetic instrumental variables; OR: odds ratio; CI: confidence interval; IVW: inverse-variance weighted; MR-PRESSO: Mendelian Randomization Pleiotropy RESidual Sum and Outlier. For HDL-C and LDL-C levels, to convert the per 10 mg/dL unit to per 1 mmol/L, the calculation for associations is exp(log(OR) × 1.8018). Bold font indicates statistical significance at *p* < 0.05.

**Table 2 genes-15-01126-t002:** Multivariable two-sample Mendelian randomization of the association between blood pressure, lipids, and HF.

Exposure	HFrEF		HFpEF	
	N GIVs	OR (95% CI)	N GIVs	OR (95% CI)
SBP (per 10 mmHg)	112	0.99 (0.88, 1.11)	111	**1.14 (1.04, 1.23)**
DBP (per 10 mmHg)	103	**1.43 (1.13, 1.77)**	104	1.05 (0.90, 1.19)
HDL-C (per 10 mg/dL)	175	0.98 (0.89, 1.02)	172	0.96 (0.90, 1.01)
LDL-C (per 10 mg/dL)	159	**1.10 (1.05, 1.17)**	159	**1.05 (1.02, 1.09)**
Triglycerides (per 10%)	175	0.99 (0.96, 1.01)	175	0.99 (0.97, 1.01)

SBP: systolic blood pressure; DBP: diastolic blood pressure; HDL-C: high-density lipoprotein cholesterol; LDL-C: low-density lipoprotein cholesterol; GIV: genetic instrumental variables; OR: odds ratio; CI: confidence interval. For HDL-C and LDL-C levels, to convert the per 10 mg/dL unit to per 1 mmol/L, the calculation for associations is exp(log(OR) × 1.8018). Bold font indicates statistical significance.

## Data Availability

The UK Biobank will make the data available to all bona fide researchers for all types of health-related research that is in the public interest, without preferential or exclusive access for any persons. All researchers will be subject to the same application process and approval criteria as specified by the UK Biobank. For more details about the procedure to access the data, see the UK Biobank website: www.ukbiobank.ac.uk. Due to U.S. Department of Veterans Affairs (VA) regulations and our ethics agreements, the analytic datasets used for this study are not permitted to leave the MVP research environment and VA firewall. However, the MVP data are available to researchers with an approved VA and MVP study protocol. The data in the BioVU cannot be shared publicly. Data are available from the Vanderbilt University Medical Center’s Synthetic Derivative and BioVU DNA BioBank for researchers who meet the criteria for access to the confidential data. The full summary level genome-wide association meta-analysis in the MVP and the BioVU will be available through dbGaP. The summary statistics for the SNPs used in the MR are available in the Appendix A.

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
