# Peer review of "A Multivariable Mendelian Randomization Study of Systolic and Diastolic Blood Pressure, Lipid Profile, and Heart Failure Subtypes"

_genes, 2024, doi:10.3390/genes15091126_

Round 1

Reviewer 1 Report

Comments and Suggestions for Authors

This is a great study. In this study, the authors showed distinct effects of blood pressure components and lipid levels on HF subtypes. The findings are of great importance for clinical practice.

I made minor revisions before accepting the manuscript for publication.

The authors could also use values in mmol/L for HDL, LDL and triglycerides.

The author could use also HR for the association assessment.

Lines 177: “This MR study investigated the origins and various mechanisms behind different 177 subtypes of HF”. I do not agree with the authors. The study did not investigate origins or mechanisms.

Author Response

Reviewer #1

This is a great study. In this study, the authors showed distinct effects of blood pressure components and lipid levels on HF subtypes. The findings are of great importance for clinical practice.

I made minor revisions before accepting the manuscript for publication.

Response: We appreciate the reviewer’s time and input.

The authors could also use values in mmol/L for HDL, LDL and triglycerides.

Response: Thank you for this comment. We chose to use the unit of mg/dL for the lipids, since it is the unit officially used by The American College of Cardiology (ACC) and the American Heart Association (AHA) (PMID: 30565953). To assist with conversions, we have included footnotes in Table 1 and Table 2 explaining how to convert from mg/dL to mmol/L, as shown below. This only applies to HDL-C and LDL-C, since the unit for triglycerides is per 10%.

Table 1. Univariable Two-sample Mendelian Randomization of The Association Between Blood Pressure, Lipids, and HF.

SBP: Systolic Blood Pressure; DBP: Diastolic Blood Pressure; HDL: High-density Lipoprotein Cholesterol; LDL: Low-density Lipoprotein Cholesterol; GIV: Genetic Instrumental Variables. OR: Odds Ratio; CI: Confidence Interval; IVW: Inverse Variance Weighted; MR-PRESSO: Mendelian Randomization Pleiotropy RESidual Sum and Outlier.

For HDL-C and LDL-C levels, to convert the per 10 mg/dL unit to per 1 mmol/L, the calculation for associations is exp(log(OR) × 1.8018).

Table 2. Multivariable Two-sample Mendelian Randomization of The Association Between Blood Pressure, Lipids, and HF.

SBP: Systolic Blood Pressure; DBP: Diastolic Blood Pressure; HDL: High-density Lipoprotein Cholesterol; LDL: Low-density Lipoprotein Cholesterol; GIV: Genetic Instrumental Variables. OR: Odds Ratio; CI: Confidence Interval.

For HDL-C and LDL-C levels, to convert the per 10 mg/dL unit to per 1 mmol/L, the calculation for associations is exp(log(OR) × 1.8018).

The author could use also HR for the association assessment.

Response: Hazard ratios (HR) are commonly used in Cox proportional hazards models with survival time-to-event data. However, our study employs Mendelian Randomization (MR) based on GWAS summary statistics derived from logistic regression models, which account for heart failure occurrences at any time and do not involve survival time data. As a result, we used odds ratios (OR) rather than HRs in our analysis. I hope that the reviewer agrees that such measure of association is appropriate for the MR study design.

Lines 177: “This MR study investigated the origins and various mechanisms behind different 177 subtypes of HF”. I do not agree with the authors. The study did not investigate origins or mechanisms.

Response: We appreciate the reviewer’s comment. At the beginning of the Discussion section, we have revised this sentence to better summarize this study as:

“This study used the MR framework to evaluate the independent causal effects of blood pressure and lipid profiles on HF subtypes.”

Reviewer 2 Report

Comments and Suggestions for Authors

The study by Liu et al. proposes a multivariable Mendelian randomization to address heart failure subtypes. Cardiovascular diseases are one of the main areas of Mendelian randomization research. Hence, I believe the authors should have explored their results better:

1. The SNPs selected are not presented, as the level of evidence, strength, fitness, and direction of the effect.

2. Although the last paragraph of the Introduction presents the analysis performed, the aim of the study could be better detailed in this section.

3.  The Discussion section is too short, more literature references should be added, especially since MR is widely used in cardiovascular diseases.

Author Response

Reviewer #2

The study by Liu et al. proposes a multivariable Mendelian randomization to address heart failure subtypes. Cardiovascular diseases are one of the main areas of Mendelian randomization research. Hence, I believe the authors should have explored their results better:

  1. The SNPs selected are not presented, as the level of evidence, strength, fitness, and direction of the effect.

Response: We agree with the reviewer that making the summaries available for the SNPs used as instrumental variables are important for research transparency and reproducibility. We have made these available in supplementary tables 2-7, which include the SNP IDs and coordinates, gene annotations, effect allele with frequency, and the beta coefficients / odds ratio with confidence intervals and p values for the exposures (blood pressure and lipids) and outcomes (heart failure with reduced and preserved ejection fraction).

  1. Although the last paragraph of the Introduction presents the analysis performed, the aim of the study could be better detailed in this section.

Response: Thank you for this suggestion. We have revised the last paragraph of the Introduction section to explain the aim of this study, as shown below:

“In this context, our study aims to provide a comprehensive evaluation of the independent causal effects of blood pressure and lipid profiles on the risk of HFrEF and HFpEF. Specifically, we utilize a multivariable two-sample Mendelian Randomization (MR) design to investigate the causal relationships between systolic blood pressure (SBP), diastolic blood pressure (DBP), pulse pressure (PP), high-density lipoprotein cholesterol (HDL-C), low-density lipoprotein cholesterol (LDL-C), and triglycerides with HFrEF and HFpEF. Our hypothesis is that these correlated factors may exert independent causal effects on HF subtypes, potentially differing in magnitude due to the distinct pathophysiological mechanisms underlying HFrEF and HFpEF. By elucidating these causal relationships, we aim to enhance the understanding of HF etiology and contribute to more targeted prevention and intervention strategies through common risk factors of HF. It is particularly important for HFpEF, which has very limited treatment options.”

  1. The Discussion section is too short, more literature references should be added, especially since MR is widely used in cardiovascular diseases.

Response: Thank you for this comment. In the Discussion section, we have added more literature review on the previous MR studies focusing on heart failure, as shown below. To further expand the Discussion section, we have emphasized the complexity of the mechanism of HF subtypes, as shown in the response to Reviewer #3.

“This study used the MR framework to evaluate the independent causal effects of blood pressure and lipid profiles on HF subtypes. Earlier inquiries had mostly concentrated on mechanisms and potential druggable targets on HF as a composite phenotype (PMID: 37429843, 31919418, 35300523, 32682105, 37935836), a limitation attributed to the absence of extensive GWAS of HF subtypes that is necessary for identifying reliable genetic instrumental variables. A recent MR study on HFrEF and HFpEF, conducted following the first GWAS on HF subtypes, relied solely on a univariable MR framework (PMID: 36517512). Our study advances this by employing multivariable MR analyses, which disentangle the independent and distinct roles of blood pressure and lipid profiles in the etiology of HF subtypes. The differences observed between univariable and multivariable MR results underscore the importance of accounting for potential pleiotropic effects that may be missed in traditional univariable MR analyses.”

Reviewer 3 Report

Comments and Suggestions for Authors

Dear Authors I would like to express tgat this is well-presented and interesting study on the different relationship between systolic and diastolic blood pressure on heart failure presentation, either as HF with reduced or preserved ejection fraction. I believe this is important conclusion from this study. The potential mechanisms of different impact of SBP and DBP should be discussed more in depth. E.g. , in patients with renal artery stenosis who suffer from renovascular hypertension and left ventricle enlargement, after renal artery stenting , patients experienced left ventricule mass reduction. However LVM reduction was independent of SBP or DBP change (https://doi.org/10.1016/j.jvs.2010.09.054 )

This SBP and DBP diversity in clinical practice was also noted in the other clinical contexts. The potential mechanisms of different impact of SBP and DBP, e.g hormonal, pre and afterload, ventricular stiffness should be mentioned in this interesting study. 

Author Response

Reviewer #3

Dear Authors I would like to express that this is well-presented and interesting study on the different relationship between systolic and diastolic blood pressure on heart failure presentation, either as HF with reduced or preserved ejection fraction. I believe this is important conclusion from this study. The potential mechanisms of different impact of SBP and DBP should be discussed more in depth. E.g. , in patients with renal artery stenosis who suffer from renovascular hypertension and left ventricle enlargement, after renal artery stenting , patients experienced left ventricule mass reduction. However LVM reduction was independent of SBP or DBP change (https://doi.org/10.1016/j.jvs.2010.09.054 )

This SBP and DBP diversity in clinical practice was also noted in the other clinical contexts. The potential mechanisms of different impact of SBP and DBP, e.g hormonal, pre and afterload, ventricular stiffness should be mentioned in this interesting study.

Response: Thank you for your insightful feedback. We greatly appreciate your recognition of the importance of our research on the differing impacts of SBP and DBP on HF presentations, particularly in the context of clinical trials that investigated the change of blood pressures and change in left ventricular mass, which is a critical mechanism of HFpEF. We have added the following statements in the Discussion section to highlight the complexity of these relationships, as shown below:

“Notably, an increase in SBP showed potential causal effect on HFpEF but not HFrEF, which is consistent with previous longitudinal study showing the stronger association between SBP and HFpEF compared with HFrEF (PMID: 34528757). SBP primarily reflects the pressure the heart works against during systole, which directly influences the stress and strain on the left ventricle. In contrast, DBP reflects the pressure during diastole, influencing the heart's filling pressure and ventricular preload. The most common cause of HFpEF is left ventricular diastolic dysfunction, caused by longstanding hypertension, especially elevated SBP, that leads to hypertrophy of the left ventricle (PMID: 9508155). The chronic pressure overload causes the left ventricle to increase in muscle mass, and this can lead to myocardial ischemia, which occurs when the heart muscle does not receive enough blood flow. The resulting fibrosis and stiffness of the left ventricle can further impair its ability of blood fill (PMID: 31256507).

In clinical trials of hypertension treatment, the relationship between reductions in left ventricular mass and changes in blood pressure values remains contentious. In the studies by Rzeznik et al. and Zeller et al., (PMID: 21129903, 17488176), left ventricular mass reduction induced by renal artery stenting or percutaneous transluminal renal angioplasty was independent of the change in blood pressure, whereas the study by Yoshitomi et al. (PMID: 8698432) reported correlations between the changes in mean blood pressure and left ventricular mass index (r = 0.77, p < 0.01) in patients with primary aldosteronism and unilateral renovascular hypertension after treatment. Interestingly, another trial of antihypertensive treatment by Rowlands et al. (PMID: 6121138) suggested that SBP has a stronger correlation with left ventricular mass compared to DBP. In addition, other studies have demonstrated that both left ventricular mass and SBP decrease in parallel during antihypertensive treatment in the context of left ventricular hypertrophy (PMID: 15547162, 28512184). Collectively, the growing evidence suggests that SBP may play a causal role in the development of HFpEF through its impact on increasing left ventricular mass. Furthermore, the hormonal mechanisms, such as those involving aldosterone may contribute to inflammation, fibrosis and stiffness of the heart (PMID: 12414524, 23774812). Conversely, DBP showed a significant MR association only on HFrEF, which may have been due to its association with an increased risk of CAD and myocardial infarction, a known contributor to left ventricular contractile dysfunction and HFrEF. These distinctions highlight the need for a nuanced understanding of the contributions of blood pressure components and specific treatment regimen to different HF subtypes, focusing on blood circulation through the heart.”